# Effect of psychoeducation in reducing caregiver burden among family caregivers of people with mental disorders in Western Ethiopia, 2021–2022: *A pre-test post-test study*

Adamu Kenea[1,2]*, Sudhakar N. Morankar[1]

1 Department of Health, Behavior and Society, College of Public Health and Medical Sciences, Jimma University, Jimma, Ethiopia, 2 Department of Psychiatry, College of Health Sciences, Mattu University, Mattu, Ethiopia

* adamukenea@gmail.com

## Abstract

### Background

Family caregivers of individuals with mental disorders experience significant financial, emotional, and psychological burden, particularly in low-resource settings with limited access to formal mental health services. Although psychoeducation is an effective strategy for reducing caregiver burden, evidence from low- and middle-income countries, including Ethiopia, remains scarce.

### Objective

To assess the effectiveness of a structured psychoeducation intervention in reducing caregiver burden among family caregivers of individuals with mental disorders attending Nekemte Comprehensive Specialized Hospital, Nekemte Town, and Mettu Karl Comprehensive Specialized Hospital, Mettu Town, Southwest Ethiopia.

### Methods

A pretest–posttest quasi-experimental study with an intervention and a comparison group was conducted from March 2021 to February 2022 at Mettu Karl Comprehensive Specialized Hospital and Nekemte Comprehensive Specialized Hospital in Oromia, Ethiopia. The sample size was determined using G*Power software, yielding a total of 556 family caregivers (intervention: n = 279; comparison: n = 277). Family caregivers were systematically sampled, with Mettu Karl Comprehensive Specialized Hospital assigned to the intervention group and Nekemte Comprehensive Specialized Hospital to the comparison group, which received routine care. The intervention consisted of six monthly structured group psychoeducation sessions. Caregiver burden was measured at baseline and post-intervention using the Zarit Burden Interview.

**Data availability statement:** All relevant data are within the paper and its supporting information files.

**Funding:** This study was supported by Jimma University through PhD dissertation funding (to A.K.). No specific funding was provided for the implementation of the intervention. The University had no role in data collection, analysis, or interpretation of the findings, decision to publish, or preparation of the manuscript.

**Competing interests:** The authors declare that there is no competing interest.

The data were entered into SPSS version 25 for analysis. Intervention effects were assessed using linear mixed-effects models and difference-in-differences analysis. Ethical approval was obtained from the Institutional Review Board of Jimma University.

## Results

There was a significant difference in mean caregiver burden scores between the intervention and comparison groups ($p < 0.001$). The intervention group demonstrated a significantly greater reduction in burden compared to the comparison group (DID = −4.32; 95% CI: −6.83, −1.81). After adjusting for socio-demographic and clinical factors, the linear mixed-effects model showed a significant reduction in caregiver burden among the intervention group ($\beta$ = −5.65; 95% CI: −7.35, −3.95; $p < 0.001$).

## Conclusion

The structured psychoeducation intervention was effective in reducing caregiver burden among families of individuals with mental disorders in Southwest Ethiopia. These findings support the integration of culturally appropriate psychoeducation programs into routine and community-based mental health services, particularly in resource-limited settings where family caregivers play a central role in patient care.

## Introduction

Mental disorders represent a major and growing public health challenge worldwide, with a disproportionately heavy impact in low- and middle-income countries (LMICs) [1]. According to the World Health Organization (WHO), mental illnesses cause hundreds of millions of disability-adjusted life years (DALYs) and millions of deaths worldwide, making them a significant contributor to premature mortality and disability [2,3]. These conditions impose far-reaching health, social, and economic consequences, making them a critical priority for health systems and development agendas.

Family care is crucial to the long-term care of people with mental disorders in resource-constrained settings like Ethiopia, where mental health services are scarce [4,5]. While this involvement is essential, it often results in considerable caregiver burden, a multidimensional construct encompassing the objective and subjective challenges of care. Objective burden refers to tangible demands such as time constraints, financial strain, and role disruption, while subjective burden captures the emotional and psychological distress associated with caregiving [6,7]. Evidence indicates that high caregiver burden is associated with depression, anxiety, social isolation, and reduced quality of life among caregivers, as well as poorer treatment adherence and outcomes for patients [8,9].

Several factors influence the level of caregiver burden. Illness-related characteristics such as symptom severity, chronicity, and functional impairment have been linked to increased stress, with caregivers of patients experiencing psychotic symptoms

or recurrent crises reporting up to threefold higher burden [10]. Health system limitations, including fragmented services, minimal family inclusion in treatment planning, and lack of structured support, further exacerbate caregiver strain [11,12]. Socioeconomic and cultural factors such as poverty, stigma, and reliance on traditional explanatory models shape caregivers' experiences and coping strategies [13–15]. In Ethiopia, more than 60% of families affected by mental illness face catastrophic health expenditures, amplifying the financial component of caregiver burden [14].

Structured psychoeducation has been recommended as an evidence-based intervention to address these challenges [16,17]. Psychoeducation typically involves providing accurate information about mental illness, training in coping and communication skills, strategies for relapse prevention, and linkage to social support. Mechanistically, such programs aim to improve illness literacy, reduce misinformation, enhance problem-solving abilities, and strengthen family resilience, thereby mitigating both objective and subjective burden [18–20]. Evidence from high-income settings demonstrates that psychoeducation reduces caregiver distress and perceived burden while improving patient adherence and relapse rates [18]. However, most evidence originates from high-resource contexts.

Caregiver-focused interventions are rarely incorporated into mental health services in Ethiopia, despite the crucial role that families play in providing care. The impact of systematic psychoeducation on caregiver burden in Ethiopia has not been extensively studied. In order to close this gap, the current study intended to assess how a multi-session psychoeducation program affects caregivers' burden among family members of individuals with mental disorders at Nekemte Comprehensive Specialized Hospital (NCSH) and Mettu Karl Comprehensive Specialized Hospital (MKCSH) in southwest Ethiopia. Assessing how caregivers' perceptions of mental illness and psychological distress have changed as a result of the intervention is one of the secondary goals.

## Methods

### Study design

A pretest-posttest non-equivalent quasi-experimental study was conducted to assess the impact of structured family psychoeducation (FPE) on burden among family caregivers (the main outcome), including comparison and intervention groups. Since random assignment was not possible, family caregivers were first enrolled in the comparison group and then in the intervention group to reduce the risk of contamination. Pre-test and post- tests were completed by both groups.

### Study setting and period

The study was conducted at Mettu Karl Comprehensive Specialized Hospital in Mettu Town, Southwest Ethiopia, and Nekemte Comprehensive Specialized Hospital in Nekemete Town, Western Ethiopia that provide inpatient and outpatient psychiatric care. The study was conducted between March 2021 and February 2022 among the family caregivers.

### Study population

The source population comprised family caregivers of patients diagnosed with any mental disorders receiving treatment at the Mettu Karl Comprehensive Specialized Hospital and Nekemete Comprehensive Specialized Hospital. A caregiver was defined as an adult family member who had the most frequent contact with the patient, provided financial or emotional support, and regularly participated in treatment or follow-up care.

### Eligibility criteria

**Inclusion criteria.** Caregivers who have provided care for at least six months were included to ensure that participants had sufficient exposure to the daily responsibilities and challenges of caregiving. This duration allows the study to capture meaningful baseline burden and ensures that the intervention addresses caregivers who are actively involved and likely to benefit from psychoeducation. Only one primary caregiver per patient was enrolled to avoid clustering effects and

duplication of data, which could bias burden assessments. This approach ensures that each patient contributes a single, independent caregiver perspective, simplifying analysis and interpretation of the intervention's effect.

**Criteria for exclusion.** Exclusion criteria included missing more than one psychoeducation session (for those in the intervention group), experiencing significant disruptive life events like Hospitalization or bereavement during the study period, and having a severe physical illness or cognitive impairment that would prevent active participation.

## Sample size and sampling procedure

G*Power version 3.1 was used to determine the sample size for a two-tailed independent-samples t test comparing the intervention and Comparison groups. No previous studies with a similar design were available to influence the estimate; Cohen's conventional benchmarks were used to assume a small-to-medium effect size (Cohen's d = 0.25). A significance level of $\alpha = 0.05$ (two-tailed) and a statistical power of 80% ($1 - \beta = 0.80$) were used to minimize the likelihood of Type I and Type II errors. With these presumptions, the required analyzable sample size is 252 individuals in each group (504 participants overall). To account for potential participant loss during follow-up (e.g., withdrawal, missing data), a 10% attrition rate was estimated. Consequently, the new ultimate recruiting target was 280 individuals each group. This adjustment ensures that the study maintains adequate power even in the presence of missing data.

Two comprehensive specialized Hospitals, Mettu Karl Comprehensive Specialized Hospital (MKCSH) and Nekemte Comprehensive Specialized Hospital (NCSH) were purposively selected for this quasi-experimental study because both institutions provide psychiatric care services and have a high volume of mental health service users. These Hospitals served as study sites for the intervention and control groups, respectively. Eligible participants were family caregivers of individuals with clinically diagnosed mental disorders who were attending outpatient psychiatric clinics at the selected Hospitals. Caregivers were included if they were aged 18 years or older, identified as the primary caregiver, and had been in the caregiving role for at least six months. From each site, eligible caregivers were identified using the outpatient psychiatric registry, which records all patients attending follow-up care. Patients diagnosed with severe mental disorders, such as schizophrenia, bipolar disorder, or major depressive disorder, were identified from the registry, and their listed primary caregivers were then approached for participation.

A systematic random sampling technique was employed to select participants from the eligible pool in both the intervention and control groups, helping to ensure representativeness and minimize selection bias. Participants from MKCSH received a structured psychoeducation intervention, while those from NCSH continued with routine psychiatric care.

## Allocation to the study arm and blinding

The study participants were assigned to either the intervention or comparison group based on the Hospital where they received care. Mettu Karl Comprehensive Specialized Hospital served as the intervention site, where structured family psychoeducation was introduced. Nekemte Comprehensive Specialized Hospital served as the Comparison site, continuing with usual care practices. A total of 277 caregivers were enrolled in the intervention group and 279 in the comparison group. The two Hospitals had comparable patient flows and staffing levels, which helped to guarantee a fair comparison and reduce overlap between the research groups even if random allocation was not feasible. Participants and intervention implementers were not blinded to the allocation because of the nature of the intervention. Interventions that involve behavior change education make it difficult to blind the providers or the receivers of educational interventions [21] (Fig 1).

## Intervention: Family Psych education (FPE)

The Family Psychoeducation (FPE) intervention used for this study was adapted from a structured model developed by Sharif et al. 2012 [22], with modifications to suit the cultural, linguistic, and practical context of family caregivers in Ethiopia. The program was delivered in small groups of eight to ten caregivers by trained mental health professionals

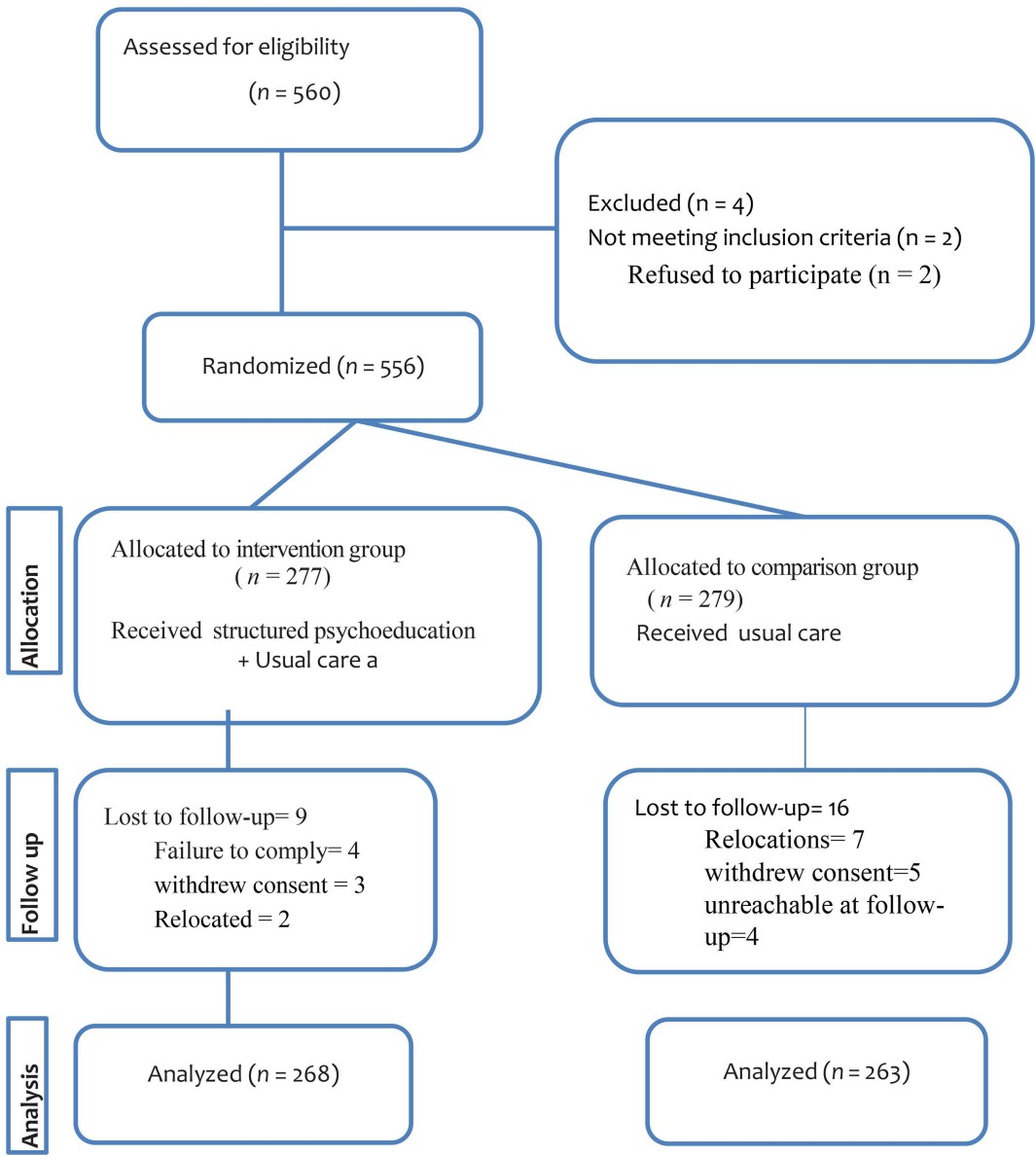

**Fig 1. Flow chart of the progress of the study participants through the phases of the intervention.**

(psychiatric nurses and psychologists). Sessions were conducted monthly over six sessions, each lasting 90–120 minutes, and delivered in the local language (Afaan Oromo) to ensure cultural and linguistic accessibility.

Each session followed a structured format: approximately 20 minutes for orientation, 60–80 minutes for content delivery, and 10 minutes for summarization. The core curriculum included topics such as understanding mental illness and local beliefs, medication adherence, symptom management, communication and problem-solving skills, stress reduction, caregiver self-care, and stigma mitigation. Interactive methods, including culturally appropriate handouts, role-plays, case discussions, and guided conversations, were used to enhance engagement and practical application. Attendance was tracked for all sessions (Table 1).

**Table 1.  The Structured Psychoeducation Intervention (Adapted from Sharif et al.2012) [22].**

| Session | Objective | Session Content |
| --- | --- | --- |
| 1 | To introduce caregivers to the intervention and build rapport | • Identification of caregiver priorities and informational needs<br>• Orientation to the structure and purpose of the program<br>• Presentation by a clinically stable patient with mental disorders sharing personal experiences<br>• Overview of common symptoms and their impact on family dynamics<br>• Discussion on the causes, course, and treatment of mental disorders<br>• Early warning signs of relapse and the family's role in relapse prevention and response |
| 2 | To enhance understanding of psychotropic medication and treatment adherence | • Recap of prior session<br>• Exploration of benefits and side effects of antipsychotic medication<br>• Emphasis on consistent medication adherence and its impact on patient stability<br>• Interactive discussion addressing caregiver concerns, myths, and barriers to adherence |
| 3 | To develop effective family communication and emotional expression | • Summary of previous content<br>• The role of family communication in mental illness recovery and relapse<br>• Instruction in verbal and non-verbal communication techniques<br>• Managing and expressing intense emotions in caregiving contexts<br>• Strategies to address emotionally charged home environments<br>• Practical discussion on dealing with patients' negative emotions |
| 4 | To build caregiver skills for managing symptoms and behavior | • Reinforcement of previous communication strategies<br>• Techniques for supporting patients during active symptoms<br>• Introduction to behavior management approaches such as token economy and reinforcement principles<br>• Skill-building exercises for responding to psychotic symptoms, withdrawal, and agitation |
| 5 | To promote caregiver self-care and stress regulation | • Overview of caregiver stress and its consequences<br>• Strategies for managing caregiving-related stress within the family setting<br>• Introduction to relaxation techniques (e.g., breathing, muscle relaxation)<br>• Guided practice and group discussion to share experiences and solutions |
| 6 | To consolidate knowledge and practice relaxation strategies | • Comprehensive review of all previous sessions<br>• Reinforcement of core concepts and practical tools<br>• Group-led relaxation practice and stress relief activities<br>• Closing reflections and feedback collection from participants |

The structured psychoeducation sessions were conducted in a quiet, well-ventilated room within each hospital, separate from routine clinical areas to minimize interruptions. Seating was arranged in a semi-circle to facilitate group interaction, discussion, and peer support. Adequate lighting and a comfortable temperature were maintained throughout the sessions. Audio-visual materials, including flip charts, posters, and handouts, were available to support learning. The room was equipped to ensure privacy, allowing participants to share experiences freely without being overheard by other patients or staff. The six-monthly psychoeducation sessions were scheduled to coincide with participants' routine outpatient visits to minimize additional travel and time burden. This approach facilitated attendance, ensured continuity of care, and allowed caregivers to integrate the sessions into their existing hospital routines. Appointment reminders were provided in advance, and session timing was coordinated with hospital staff to avoid conflicts with clinical activities. To reduce information contamination, the intervention and comparison groups were recruited from different hospitals, ensuring that caregivers in the comparison group did not have access to or participate in the psychoeducation sessions during the study period.

### Intervention fidelity

The intervention fidelity was evaluated using best practice recommendations developed by the National Institutes of Health (NIH) Behavioral Change Consortium [23]. The recommendations cover (i) study design: establishing procedures to monitor and reduce the potential for contamination between active treatments or treatment and control, as well as to

measure dose and intensity; (ii) provider training: standardization of training to ensure that all providers are trained in the same manner; (iii) treatment delivery: behavioral checklists to ensure that providers adhered to the treatment protocol and (iv) treatment receipt: through supervisory visits to the study area and performance evaluation meetings with the Psycho-education providers.

## Data collection tools and procedures

Data were collected through face-to-face interviews by trained psychiatric nurses not involved in intervention delivery. Baseline assessments were conducted before intervention sessions, and follow-up evaluations were completed post-intervention after 6 months for the intervention group.

The primary outcome was caregiver burden, assessed using the 22-item Zarit Burden Interview (ZBI) [24,25] baseline and immediately post-intervention for both groups. The ZBI uses a Likert scale, with higher scores indicating greater caregiver burden. Other variables collected included socio-demographic characteristics (age, sex, education, marital status, economic status, relationship to the patient, and duration of caregiving) and patient clinical information (Supporting Information 1 in S1 Data).

The Beliefs toward Mental Illness Scale (BMIS) is a 21-item [26] instrument designed to assess attitudes towards mental illness and its treatment. The BMIS utilizes a three-point Likert response format: Disagree (Negative), Neutral (Undecided), and Agree (Positive). Questionnaire items were adapted from the original BMIS tool and supplemented with insights from relevant literature.

## Data management and analysis

Data were double-entered in EpiData v3.1 and exported to SPSS version 23.0 for analysis. Descriptive statistics, including means, standard deviations, and frequencies, summarized socio-demographic and clinical characteristics. Between-group differences in post-intervention caregiver burden were analyzed using a linear mixed-effects model (LMM), adjusting for baseline scores. Within-group changes were assessed using paired t-tests or Wilcoxon signed-rank tests, as appropriate. The study population was described using summary statistics of means and percentages based on the study outcomes, socio-demographic characteristics, and other factors. To estimate the average treatment effect on the treated, a difference-in-differences (DID) approach was employed. This method compares the changes in outcome measures over time between the intervention and comparison groups. By accounting for temporal trends and group-specific characteristics that are not directly observed, DID helps isolate the true effect of the intervention from potential confounding influences.

To evaluate the effect of the intervention over time while accounting for repeated measures, a linear mixed-effects model (LMM) was employed. The model included fixed effects for time (baseline, post-intervention), group (intervention vs. comparison), and the time × group interaction, which represented the primary test of the intervention effect. Random intercepts for participants were included to account for individual-level variability. This approach accommodates correlated observations, unbalanced data, and missing values under the assumption of missing at random (MAR). Model assumptions of normality of residuals, homoscedasticity, and appropriate covariance structure were assessed through residual diagnostics, and alternative covariance structures were explored to ensure the best fit based on Akaike Information Criterion (AIC). The normality assumption of the outcome variable (burden) was assessed by using the Shapiro−Wilk test, and the test showed that the assumption was satisfied ($p > 0.05$). We used the Akaike information criterion (AIC) to assist in selecting the appropriate statistical model. We chose the model that demonstrated the minimum AIC. Variables with $p < 0.2$ in the bivariate linear mixed regression model were selected as candidate variables for the multivariable linear mixed model analysis. The intervention's effectiveness was assessed by examining the interaction between time and the intervention. Effect sizes were expressed as mean differences with 95% confidence intervals, and statistical significance was set at $p < 0.05$.

 

**Patient and public involvement**

Patients were not involved in the design, conduct, reporting, or dissemination plans of this research. Study findings will be shared with participants and the wider public through study reports and publication in an open-access journal.

**Ethics approval and consent to participate**

The study protocol was reviewed and approved by the Institutional Review Board (IRB) of Jimma University (IRB approval no.: IHRPGn/944/20). Administrative permission was obtained from each participating hospital prior to data collection. All eligible caregivers were informed about the study's purpose, procedures, potential benefits, and minimal risks, and questions were encouraged. Informed written consent was obtained in the participant's preferred language after confirming comprehension, considering varying literacy levels. Participation was entirely voluntary, and caregivers were assured that refusal or withdrawal at any time would not affect the care they or their family member received. Confidentiality was maintained by assigning unique study IDs, removing personal identifiers from analytic datasets, and storing data in password-protected files accessible only to the research team. As a benefit, a condensed psychoeducation session was provided to the comparison group caregivers after study completion. The study adhered to the ethical principles of the Declaration of Helsinki for research involving human subjects [27].

## Result

### Socio-demographic characteristics of caregivers

The flow of the study participants through the intervention process is depicted in Fig 1. At baseline, the intervention and comparison groups were largely similar across all the socio-demographics, suggesting general comparability. No statistically significant differences were observed in age (p = 0.20), sex (p = 0.106), residence (p = 0.664), marital status (p = 0.864), religion (p = 0.756), educational level (p = 0.166), employment status (p = 0.772), family size (p = 0.317), or patient diagnosis (p = 0.105). However, a notable difference was found in the relationship to the patient. A statistically significant difference was observed in the relationship with the patient (p = 0.032). Caregivers in the intervention group were more likely to be siblings (21.7%) or cousins (13.0%), whereas the comparison group had a higher proportion of siblings (31.2%) and spouses (13.0%) (Table 2).

### Caregivers' burden

The findings of this study showed that the mean caregiver burden scores were comparable between the intervention and comparison groups at baseline, with no statistically significant difference (33.31 ± 11.87 vs. 34.47 ± 12.03). However, at the end line, both groups exhibited a reduction in mean burden scores, with the intervention group showing a more substantial decline (26.77 ± 7.69) compared to the comparison group (32.25 ± 10.51). The mean difference from baseline to end line was −6.54 (95% CI: −8.21, −4.87) for the intervention group and −2.22 (95% CI: −4.09, −0.35) for the comparison group. The difference-in-differences (DID) analysis revealed a statistically significant net reduction in caregiver burden in the intervention group relative to the comparison group (DID = −4.32; 95% CI: −6.83, −1.81). The differences were statistically significant (p < 0.05) (Table 3).

### Effect of the Psychoeducation on burden among the family caregivers

The intra-individual correlation coefficient (ICC) in the null model was 0.356, indicating that approximately 36% of the total variance in caregiver burden was attributable to differences between subjects, highlighting the importance of accounting for individual-level time-invariant variables using a two-level model. After controlling for socio-demographic factors including caregivers' age, educational status, employment status, family size, marital status, relationship with the patient, and baseline attitude towards mental illness caregivers in the intervention group exhibited a substantial reduction in burden

**Table 2. Baseline socio-demographic and Clinical Characteristics of Respondents by Group, MKCSH and NCSH, Western Ethiopia, 2021 - 2022.**

| Variable | Category | Intervention (n = 277) N (%) | Comparison (n = 279) N (%) | p-value |
|---|---|---|---|---|
| Age | Mean (±SD) | 31.91(11.12) | 33.14(11.50) | 0.20 |
| Sex | Male | 137 (49.5) | 119(42.7) | 0.106 |
|  | Female | 140 (50.5) | 160 (57.3) |  |
| Place of residence | Rural | 168 (60.6) | 164 (58.8) | 0.664 |
|  | Urban | 109 (39.4.3) | 115 (41.2) |  |
| Marital Status | Single | 154 (55.6) | 147 (52.7) | 0.864 |
|  | Married | 108 (39.0) | 113 (40.5) |  |
|  | Separated | 8 (2.9) | 10 (3.6) |  |
|  | Widowed/Widower | 7 (2.5) | 9 (3.2) |  |
| Religion | Christian | 158 (57.0) | 161 (57.7) | 0.756 |
|  | Muslim | 116(41.9) | 113 (40.5) |  |
|  | Wakefata | 3 (1.1) | 5(1.8) |  |
| Ethnicity | Oromo | 233 (84.1%) | 250 (89.6%) | 0.055 |
|  | Amhara | 38 (13.7%) | 21 (7.5%) |  |
|  | Others* | 6 (2.2%) | 8 (2.9%) |  |
| Educational Level | No Formal Education | 136 (49.1) | 139 (49.8) | 0.166 |
|  | Primary Education | 34 (12.3) | 51 (18.3) |  |
|  | Secondary Education | 83 (30.0) | 69 (24.7) |  |
|  | Tertiary Education | 24 (8.7) | 20 (7.2) |  |
| Employment Status | Unemployed | 88 (31.8) | 78 (28.0) | 0.772 |
|  | Self-employed | 82 (29.6) | 87 (31.2) |  |
|  | Civil Servant | 74 (26.7) | 84 (30.1) |  |
|  | Student | 23 (8.3) | 19 (6.8) |  |
|  | Daily laborers | 10 (3.6) | 11 (3.9) |  |
| Family Size | <5 | 174 (62.4) | 105 (37.6) | 0.317 |
|  | ≥5 | 185 (66.8) | 92 (33.2) |  |
| Relationship with Patient | Parent | 108(39.0) | 101 (36.2) | 0.032 |
|  | Sibling | 60 (21.7) | 87(31.2) |  |
|  | Uncle/Aunt | 33 (11.9) | 25(9.0) |  |
|  | Cousin | 36(13.0) | 23 (8.2) |  |
|  | Spouse | 27 (9.7) | 36 (13.0) |  |
|  | Others** | 13 (4.7) | 7(2.7) |  |
| Patient Diagnosis | Mental disorders | 63(22.7) | 71 (25.4) | 0.105 |
|  | Bipolar Affective Disorder | 30 (10.8) | 33 (11.8) |  |
|  | Depression | 66 (23.8) | 80 (28.7) |  |
|  | Substance Abuse | 84 (30.3) | 59 (21.1) |  |
|  | Mental Retardation | 21(7.6) | 15 (5.4) |  |
|  | Partial Seizures | 3 (1.1) | 9 (3.2) |  |
|  | Others*** | 10 (3.6) | 12 (4.3) |  |

*Tigray, Gurage. ** Grandparent, Step-parent *** Autism Spectrum Disorder, Personality Disorders

**Table 3. Baseline, Endline, Mean Differences, and Difference-in-Differences.**

| Group | Baseline Mean (±SD) | 95% CI | Endline Mean (±SD) | 95% CI | Mean Difference (EL − BL) | 95% CI |
|---|---|---|---|---|---|---|
| Intervention Group | 33.31 (11.87) | 31.91–34.72 | 26.77 (7.69) | 25.86–27.68 | −6.54 | −8.21, −4.87 |
| Comparison Group | 34.47 (12.03) | 33.05–35.88 | 32.25 (10.51) | 31.02–33.49 | −2.22 | −4.09, −0.35 |
| Difference-in-Differences (DID) (±SE) | – | – | – | – | −4.32 (±1.28)(** | −6.83, −1.81 |

Abbreviations: BL, baseline; EL, End line; CI, confidence interval; SD, Standard deviation. **p < 0.01.

scores at the end of the study (β = −6.44; 95% CI: −8.15, −4.72; p < 0.001). The effect of time was also significant, with burden scores decreasing from baseline to endline (β = −5.65; 95% CI: −7.35, −3.95; p < 0.001), and the interaction between time and group confirmed that changes in burden scores differed significantly between intervention and comparison groups (β = −5.65; 95% CI: −7.35, −3.95; p < 0.001).

Regarding model fit, the level-two variance decreased from 33.23 in the null model to 10.42 in the fully adjusted model, while the ICC decreased slightly from 0.356 to 0.339, suggesting that the intervention and covariates explained a substantial portion of between-subject variability, though meaningful clustering remained. Model comparison based on AIC values demonstrated progressive improvement in fit across models: the null model had an AIC of 8071.96, the random-intercept model including group and time had an AIC of 7973.00, and the full model incorporating all fixed effects had the lowest AIC (7691.26) and BIC (7750.62), indicating the best overall fit. Overall, these results show that the intervention effectively reduced caregiver burden while accounting for clustering and relevant covariates, consistent with the study's primary objective (Table 4).

## Discussion

This study examined the effect of a psychoeducation intervention on the burden experienced by family caregivers of individuals with mental disorders in Western Ethiopia. The findings revealed a statistically significant reduction in caregiver burden among those who received the intervention. These results are consistent with previous studies, including one by Adisa et al. (2025) in Nigeria [28], which reported that a structured psychoeducation program led to a significant decrease in caregiver burden among families of patients with mental disorders. Similarly, a study in Iran found that participants in a psychoeducation group experienced a significant decline in caregiver burden, whereas no meaningful change was observed in the comparison group [29]. Consistent findings were also reported in Egypt, where an educational program led to a substantial reduction in burden among the majority of participating family caregivers [30]. Additionally, another study in Nigeria confirmed that caregivers receiving structured psychoeducation reported greater reductions in burden compared to those receiving usual care [31]. Further supporting this evidence, a systematic review and meta-analysis concluded that psychoeducation interventions are more effective than standard care in reducing caregiver burden among families of adults with mental disorders [32]. The consistency of these findings across diverse settings can be justified by several key mechanisms. Psychoeducation works by increasing knowledge and illness literacy, which reduces fear and helplessness; enhancing practical coping and problem-solving skills to manage daily challenges; reducing stigma and isolation through group support; and fostering caregiver self-efficacy. Furthermore, in resource-constrained contexts like Ethiopia, Nigeria, and Egypt, caregivers often face a high baseline of unmet need. Culturally adapted psychoeducation directly addresses this critical gap, providing structured support that routine care typically lacks, which explains its significant and replicable effect on alleviating burden.

However, not all studies have reported similar findings. A Japanese study by Yasuma et al. (2024) found that brief family psychoeducation did not significantly reduce caregiver stress at one or six months of follow-up [33]. The observed discrepancy could be explained by environmental factors and differences in the caregivers' burden at baseline. The

**Table 4. Linear mixed-effects model predicting the Effect of Psychoeducation on Caregiver Burden Scores among Family Caregivers of People with Mental Disorders (n = 531, I = 268, C = 263).**

| Fixed effect | Model 1 Estimate (95% CI) | Model 2 Estimate (95% CI) | Model 3 Estimate (95% CI) |
|---|---|---|---|
| Variables | | | |
| Intercept | 32.18 (31.52, 32.83)*** | 33.16 (31.90, 34.41)*** | 31.29 (23.48, 39.09)*** |
| Intervention effect | | | |
| Intervention group | – | −6.31 (−8.07, −4.55)*** | −6.44 (−8.15,-4.72,)*** |
| Comparison Group (Ref) | – | – | – |
| Intervention and Group Interaction | – | | |
| End line | – | −5.65 (−7.35,-3.95)***, | −5.65 (−7.35, −3.95)*** |
| Baseline(Ref) | – | – | – |
| Random effect | | | |
| Level two variance | 33.231 | 25.782 | 10.421 |
| AIC | 8071.96 | 7973.00 | 7691.26 |
| ICC | 0.356 | 0.357 | .339 |

***p < 0.001, **p < 0.01, *p < 0.05.

Note: Model 1. Intercept-only model; Model 2. Slope-only model; Model 3. Intercept with slope.

Abbreviations: AIC, Akaike information criteria; CI, confidence interval.

The model was adjusted for the caregivers' age, educational status, employment status, family size, marital status, relationship with the patient, and Baseline attitude.

Japanese study's smaller sample size and lower starting burden levels might have limited the intervention's discernible effects. In contrast, the present study involved caregivers with higher baseline burden, providing greater scope for improvement. Moreover, the structured psychoeducation format and a supportive delivery environment may have facilitated more effective engagement, highlighting the importance of considering baseline burden, cultural context, and intervention delivery in psychoeducation program design.

The analysis, after controlling for relevant socio-demographic factors and baseline attitudes, confirmed a substantial reduction in burden scores specifically within the intervention group. This finding underscores the targeted efficacy of the psychoeducation program. In contrast, the minimal reduction observed in the Comparison group highlights a critical limitation of routine mental health services, which appear insufficient to address the complex psychosocial and educational needs of caregivers. This disparity reinforces the value of adding structured, skill-based support to standard clinical care, particularly in settings where families are the primary and often unsupported backbone of long-term care.

This study has several methodological strengths that bolster the validity and relevance of its findings. First, the relatively large sample size (*n* = 556) enhances statistical power and improves the generalizability of results to family caregivers in similar low-resource, hospital-based settings in Ethiopia. Second, we employed validated instruments, such as the Zarit Burden Interview, which increases the reliability and comparability of the burden assessments. Third, the intervention's extended duration of six monthly sessions provided ample time for skill acquisition, group cohesion, and potential behavioral embedding, factors important for sustaining caregiver outcomes. Finally, the use of advanced analytic methods, including difference-in-differences and linear mixed-effects models, allowed us to isolate the intervention effect while adjusting for baseline differences and individual-level variability, thereby strengthening causal inference. Several limitations should be considered when interpreting the results. Primarily, all outcome data were based on self-report, which may be susceptible to social desirability bias and recall inaccuracy. Although we used validated tools, the lack of objective or observational measures of caregiver burden limits the ability to fully capture multidimensional aspects of the construct. Additionally, the quasi-experimental design, with allocation by hospital site rather than randomization, introduces the

possibility of selection bias and unmeasured confounding, despite statistical adjustments. Finally, the 6-month follow-up period, while adequate for detecting short-term changes, does not inform the longer-term sustainability of burden reduction or the intervention's impact on patient outcomes.

### Implications

**Clinical Practice Implications**: The findings support the expansion of caregiver-focused psychoeducation as a routine component of mental health care. Such interventions should use culturally appropriate materials, structured and standardized content, and be delivered by adequately trained facilitators to ensure fidelity and effectiveness. Regular assessment of caregivers' mental health literacy and caregiving burden is recommended to guide individualized support and timely referral to additional services, thereby improving both caregiver capacity and patient outcomes.

**Policy Recommendations**: From a policy perspective, the results underscore the need to formally integrate family psychoeducation into national mental health strategies and primary health care systems. Policymakers should prioritize structured support for family caregivers as a core element of psychiatric care reform, particularly in resource-limited settings where families serve as the main providers of long-term care. Allocating resources for caregiver training, supervision, and monitoring frameworks would strengthen sustainable implementation.

**Future Research Directions**: Future studies should examine the long-term effects of caregiver psychoeducation on patient clinical outcomes, caregiver well-being, and health system utilization. In addition, implementation research is needed to evaluate the feasibility, scalability, and cost-effectiveness of integrating caregiver psychoeducation into existing mental health service delivery models, including community-based and primary care settings.

## Conclusion

The findings of this study demonstrated that psychoeducation is an effective approach for alleviating the burden experienced by family caregivers of people with mental disorders s in Western Ethiopia. Participants who received the intervention reported a significant reduction in caregivers' burden, highlighting the practical value of structured and culturally relevant educational support. The results further emphasize that caregiver burden is influenced not only by caregiving responsibilities but also by broader contextual factors such as education level, household size, and marital status. These insights suggest that psychoeducation programs may be especially beneficial when targeted toward caregivers with limited support or resources. By fostering greater understanding, improving coping strategies, and providing a space for shared experience, psychoeducation can serve as a key component of caregiver support within mental health services. Integrating such interventions into routine care, while adapting them to the cultural and emotional realities of caregivers, may enhance their effectiveness and sustainability. Future research should consider evaluating the long-term impacts of psychoeducation and exploring scalable models to reach broader populations in similar settings.

## Supporting information

**S1 Data. Excell data.**
(XLS)

**S2 Data. Tools used to assess burden and Attitude.**
(DOCX)

## Acknowledgments

We would like to express our sincere gratitude to Jimma University, Mettu Karl Comprehensive Specialized Hospital (MKCSH), and Nekemte Comprehensive Specialized Hospital (NCSH) for their invaluable support in the implementation of this study. We also extend our heartfelt thanks to the data collectors, supervisors, study participants, and all others who

contributed to the successful completion of this research. The views expressed in this publication are those of the authors and do not necessarily represent the official position of the sponsoring institution.

## Author contributions

**Conceptualization:** Adamu Kenea.

**Data curation:** Adamu Kenea, Sudhakar N. Morankar.

**Formal analysis:** Adamu Kenea, Sudhakar N. Morankar.

**Funding acquisition:** Adamu Kenea, Sudhakar N. Morankar.

**Investigation:** Adamu Kenea.

**Methodology:** Adamu Kenea, Sudhakar N. Morankar.

**Project administration:** Adamu Kenea, Sudhakar N. Morankar.

**Resources:** Adamu Kenea.

**Software:** Adamu Kenea, Sudhakar N. Morankar.

**Supervision:** Adamu Kenea, Sudhakar N. Morankar.

**Validation:** Adamu Kenea, Sudhakar N. Morankar.

**Visualization:** Adamu Kenea.

**Writing – original draft:** Adamu Kenea.

**Writing – review & editing:** Adamu Kenea, Sudhakar N. Morankar.

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
