## [Decision Letter · Decision Letter 0]

22 Dec 2025

Dear Dr. Kenea,

Thank you for submitting your manuscript to PLOS ONE. After careful consideration, we feel that it has merit but does not fully meet PLOS ONE’s publication criteria as it currently stands. Therefore, we invite you to submit a revised version of the manuscript that addresses the points raised during the review process.

We look forward to receiving your revised manuscript.

Kind regards,

Zakir Abdu Adem, PhD fellow, La Trobe University

Academic Editor

PLOS One

Journal Requirements:

2. In this instance it seems there may be acceptable restrictions in place that prevent the public sharing of your minimal data. However, in line with our goal of ensuring long-term data availability to all interested researchers, PLOS’ Data Policy states that authors cannot be the sole named individuals responsible for ensuring data access (http://journals.plos.org/plosone/s/data-availability#loc-acceptable-data-sharing-methods).

Before we proceed with your manuscript, please also provide non-author contact information (phone/email/hyperlink) for a data access committee, ethics committee, or other institutional body to which data requests may be sent. If no institutional body is available to respond to requests for your minimal data, please consider if there any institutional representatives who did not collaborate in the study, and are not listed as authors on the manuscript, who would be able to hold the data and respond to external requests for data access? If so, please provide their contact information (i.e., email address). Please also provide details on how you will ensure persistent or long-term data storage and availability

Additional Editor Comments (if provided):

Title: Add the year of the study on the titleAbstract:

Rather than said Western Ethiopia, mention a specific study settingI don’t know as the name of hospitals correctly written. So, please write in the correct form by consulting the hospital administrator, and attach the letter they gave to you the correct name of hospital. Rationale: when someone search as a keywords, there will be different name of hospitalHow was the sample size was calculated?Who was your study participants? Caregivers or patients? If caregivers, what is the routine care they received?At the end of the abstract, the protocol registration is needed. If you registered, please cite. If not registered, full proposal is needed

Introduction:

LMICs – put in the bracketsfamily psychoeducation (FPE) – write the first words in caps lock

Methods:

The training was given for 90 to 120 minutes for 6 times. The participants burned many of their time. I think they should get some advantages for their time.The training or intervention was delivered by Afan Oromo. Did all participants talk same language?Specify the room where they took the treatment and how you got the room from the hospitals?

Results: needs English grammar editionDiscussion: discussion is the heart for the research. The discussion part should rewrite. It is well if you elaborate more. Rather than comparing your result with the previous findings, elaborate more by using your experiences (clinical, academics) and support by referencesWhat is the implication of the study? Well, if elaborate itGenerally, the paper needs English grammar edition

Reviewers' comments:

Reviewer's Responses to Questions

**Comments to the Author**

1. Is the manuscript technically sound, and do the data support the conclusions?

Reviewer #1: Yes

Reviewer #2: Yes

2. Has the statistical analysis been performed appropriately and rigorously?

Reviewer #1: Yes

Reviewer #2: Yes

3. Have the authors made all data underlying the findings in their manuscript fully available?

Reviewer #1: Yes

Reviewer #2: Yes

4. Is the manuscript presented in an intelligible fashion and written in standard English?

Reviewer #1: Yes

Reviewer #2: No

Reviewer #1: Dear Authors,

You have conducted an excellent and much-needed study on an important topic that has been largely neglected and under-researched in many parts of the country. Your work is well-designed, thoroughly executed, and clearly presented. The methodology is appropriate, the analysis is sound, and the conclusions are well-supported by the findings.

Overall, this manuscript makes a valuable contribution to the field.

Reviewer #2: This is a review of the manuscript “Effect of Psychoeducation in Reducing Caregiver Burden among Family Caregivers of People with Mental Disorders in Western Ethiopia: A Pre-Test Post-Test Study”. This study evaluated the effectiveness of structured psychoeducation in reducing Caregiver Burden among Family Caregivers of People with Mental Disorders in Western Ethiopia.

Authors found differences in mean burden score among both groups. The finding significantly greater reduction in burden scores. This paper addresses an important topic and novel in its nature. However, there are certain concerns related to the conceptualization, design of the study, and reporting/writing of the manuscript.

There is missing information, grammatical error, and lack of clarity in writing, making it incredibly difficult to review the manuscript and understand the impact.

See more detailed comments below:

1. It is not clear how the intervention was delivered particularly with regard to reducing the risk of information contamination at both study setting. Authors stated that…. Since random assignment was not possible, family caregivers’ were first enrolled in the comparison group and then in the intervention group…make it clear enough.

2. Study population section: Typos error : disorder

3. Exclusion criteria: duration of caring is an important for the family caregivers: it is not clear how authors managed to select them??

In addition why restriction in family: which one do you prefer? For example if they father or mother of the patient? This will cause selection bias?? ….Only one caregiver per patient was allowed to participate in the study, and …..

Why only in one afaan Oromo: ‘‘family caregivers’ also had to be able to converse in Afaan Oromo’’ …is there a logical explanation

4. Repetition of phrases: already stated. Add 6month duration in the inclusion criteria in addition of the other section…… ‘‘Caregivers were included if they were aged 18 years or older, identified as the primary caregiver, and had been in the caregiving role for at least six months……’’

5. Lacks clarity: Do you think there is well registered data for the caregivers???……From each site, eligible caregivers were identified through the outpatient psychiatric registry.’’ Again for what types of condition you have selected the caregivers as the burden vary for major and minor psychiatric condition.

6. Implementation: it is difficult to deliver this intervention without provision of incentives for the patient of caregivers?? In not how was the intervention was provided? Since they need to visit for the regular follow and also have to attend the session. This needs explanation.

7. Data analysis: typos…. Interaction

8. Tense: Study findings were shared with participants and the wider public through study reports and publication in an open access journal.

9. Sociodemographic data: Typos: sex was

10. Discussion: typos: Comparison…..

11. Conclusion: typos: … mental disorders s in Western Ethiopia.

12. I doubt the source of fund for the current study. This is large scale intervention requiring experts for the supervision. Please clarify??

.

Reviewer #1: **Yes:** Beshir MammiyoBeshir MammiyoBeshir MammiyoBeshir Mammiyo

Reviewer #2: **Yes:** Mohammedamin HajureMohammedamin HajureMohammedamin HajureMohammedamin Hajure

---

## [Author Response · Author response to Decision Letter 1]

30 Jan 2026

Point-by-Point Response to Editor and Reviewer Comments

Response to Editor Comments:

We thank the Editor for the thorough review and for the opportunity to improve our manuscript. We sincerely appreciate the specific and constructive feedback provided. We have addressed each comment below and incorporated the necessary revisions into the revised manuscript.

1. Title:

Thank you for this helpful suggestion. We have revised the title to include the year of data collection:

Effect of Psychoeducation in Reducing Caregiver Burden among Family Caregivers of People with Mental Disorders in Western Ethiopia: A Pre-Test Post-Test Study, 2021 - 2022.

2. Abstract – Study Setting:

We appreciate this important clarification. We replaced "Western Ethiopia" with the specific administrative zones and towns:

"…was conducted at Nekemte Comprehensive Specialized Hospital in Nekemte Town, East Wollega Zone and Mettu Karl Comprehensive Specialized Hospital I Ilu Aba Bor Zone, Southwest Ethiopia."

1. Abstract – Hospital Names:

Thank you for raising this critical point. We had official permission letters from each hospital's administration stating the correct, official names.

Abstract – Sample Size:

We appreciate this valuable comment. We added a sentence in the abstract stating:

The sample size was determined using G*Power software, yielding a total of 556 family caregivers (intervention: n = 279; comparison: n = 277).

2. Abstract – Study Participants:

Thank you for highlighting the need for clarity. We clarified: The study participants were primary family caregivers of patients diagnosed with severe mental disorders (schizophrenia, bipolar disorder, or major depressive disorder). Caregivers in the intervention group received a structured family psychoeducation (FPE) intervention in addition to the routine clinical care (which included medication management and brief advice from treating psychiatrists)."

3. Abstract – Protocol Registration:

We appreciate this important methodological concern. The study protocol was not prospectively registered in a public trial registry. However, the full study proposal was reviewed and approved by the Institutional Review Board (IRB) of Jimma University. We indicated the IRB approval in the methods section and stated in the abstract: The study protocol was approved by the Jimma University IRB.

3. Introduction – LMICs:

Thank you for noting this terminology issue. We wrote it as "low- and middle-income countries (LMICs)" at first use.

1. Introduction – FPE: We appreciate this clarification request. We wrote it as "Family Psychoeducation (FPE)" at first use.

2. Methods – Advantages for Participants:

Thank you for this important ethical consideration. Participants did receive benefits, which we stated more clearly:

While no direct financial incentive was provided, participants in the intervention group received structured psychoeducation, which is a valuable supportive service not routinely available. All participants, in both groups, continued to receive standard psychiatric care for their patient-relative."

4. Methods – Language:

We appreciate this important contextual comment. We clarified: The intervention was delivered in Afaan Oromo, the local language spoken by all participants and the primary language of the study region, which ensured full comprehension."

1. Methods – Intervention Room:

Thank you for requesting this clarification. We added: The group sessions were conducted in a private, quiet meeting room within the outpatient department of each hospital, which was allocated by the hospital administration for the purpose of this study

5. Results:

We appreciate this helpful observation. We performed a comprehensive English grammar edit of the entire Results section and the manuscript.

6. Discussion:

We thank the Editor for this crucial and insightful feedback. We substantially rewrote the Discussion to:

(a) more deeply interpret our findings in the context of our clinical and academic experiences in Ethiopian mental health care;

(b) elaborate on the mechanisms (e.g., improved knowledge, coping skills, social support) that might explain the results, supported by relevant references; and

(c) move beyond simple comparison to synthesize why our findings aligned or differed from previous literature.

7. Implications:

Thank you for this valuable suggestion. We expanded the "Implication" subsection to clearly distinguish between clinical practice (e.g., integrating FPE into routine care), policy (e.g., task-sharing and training non-specialists), and future research (e.g., longer-term follow-up, cost-effectiveness).

8. General Grammar:

We appreciate this overarching comment. As noted, we undertook a complete professional English language edit of the entire manuscript to ensure clarity and correctness.

Response to Reviewer #2’s Comments:

We extend our sincere gratitude for your thoughtful and detailed review of our manuscript. Your insightful comments were invaluable for strengthening our paper. We addressed each point below.

Risk of Information Contamination:

Thank you for raising this important methodological concern. To reduce information contamination, the intervention and comparison groups were recruited from different hospitals, ensuring that caregivers in the comparison group did not have access to or participate in the psychoeducation sessions during the study period.

1. Study Population – Typo:

2. Thank you for noting this error. We corrected "disorder" to "disorders."

3. Exclusion/Inclusion Criteria:

• Duration of Care: Thank you for requesting justification. We specified a minimum of six months to ensure caregivers had sustained experience of the caregiving role and clarified this rationale in the text.

• One Caregiver per Patient: We appreciate this methodological concern. This criterion ensured independent observations and prevented clustering effects, as the burden experience of multiple caregivers for one patient is not independent. We added this explanation.

• Language (Afaan Oromo): Thank you for highlighting this issue. The study was conducted in the Oromia region where Afaan Oromo is the predominant native language. This criterion ensured that participants could fully engage with the culturally and linguistically adapted intervention materials and group discussions without a language barrier. We made this rationale explicit.

4. Repetition of Phrases:

We appreciate this careful observation. We streamlined the criteria to avoid repetition, keeping the six-month duration only in the inclusion criteria section.

5. Clarity on Registry and Patient Condition:

• Registry: Thank you for seeking clarification. The outpatient psychiatric registry logged all patients attending follow-up. From this, we identified patients with the specified diagnoses and then approached their listed primary caregiver.

• Patient Diagnosis: We appreciate this conceptual comment. We focused on severe mental disorders (schizophrenia, bipolar disorder, major depressive disorder) precisely because caregiver burden is typically high and persistent in these conditions, making the intervention highly relevant. We stated this focus more clearly in the methods.

6. Intervention Delivery without Incentives:

Thank you for raising this practical concern. Participants were already attending the hospital for the patient’s routine monthly medication refills. The psychoeducation sessions were scheduled to coincide with these visits to minimize additional travel burden. We elaborated on this logistical planning in the methods.

7. Data Analysis – Typo:

We appreciate this careful review. We corrected "interaction" and all other typographical errors.

8. Tense:

Thank you for pointing this out. We corrected the sentence to the past tense:

"Study findings were shared with participants and the wider public through study reports and publication in an open access journal."

9. Socio-demographic – Typo:

Thank you for noting this error. We corrected "sex was" to "sex were."

10. Discussion – Typo:

We appreciate this correction. We corrected "Comparison."

11. Conclusion – Typo:

Thank you for identifying this issue. We corrected "disorders s" to "disorders."

12. Source of Funding and Supervision:

Thank you for raising this important transparency issue. The study was conducted as part of a PhD program. A mental health professional delivered the intervention under the regular supervision of the Principal Investigator (PhD student). No external funding was received for direct implementation costs (e.g., incentives), which was a study limitation. We added the following statement in the funding section: This research was conducted as part of a PhD thesis. No specific funding was received for the implementation of the intervention."

We sincerely thank both the Editor and Reviewer for their time, expertise, and constructive feedback. We believe these revisions have significantly enhanced the quality, clarity, and impact of our manuscript. We have submitted the revised version accordingly.

---

## [Decision Letter · Decision Letter 1]

3 Mar 2026

Effect of Psychoeducation in Reducing Caregiver Burden among Family Caregivers of People with Mental Disorders in Western Ethiopia, 2021 - 2022: A Pre-Test Post-Test Study

PONE-D-25-47275R1

Dear Mr. Kenea%,

We’re pleased to inform you that your manuscript has been judged scientifically suitable for publication and will be formally accepted for publication once it meets all outstanding technical requirements.

Kind regards,

Zakir Abdu Adem, PhD fellow

La Trobe University, Melbourne, Australia

Academic Editor

PLOS One

Additional Editor Comments (optional):

Thank you for submitting your manuscript for publication. Your work has the potential to add value to the field and may contribute meaningfully to patients and their attendants.

Reviewers' comments:

Reviewer's Responses to Questions

**Comments to the Author**

Reviewer #1: All comments have been addressed

Reviewer #2: All comments have been addressed

2. Is the manuscript technically sound, and do the data support the conclusions?

Reviewer #1: Yes

Reviewer #2: Yes

3. Has the statistical analysis been performed appropriately and rigorously?

Reviewer #1: Yes

Reviewer #2: Yes

4. Have the authors made all data underlying the findings in their manuscript fully available?

Reviewer #1: Yes

Reviewer #2: Yes

5. Is the manuscript presented in an intelligible fashion and written in standard English?

Reviewer #1: Yes

Reviewer #2: Yes

Reviewer #1: I appreciate the authors’ efforts in revising the manuscript and addressing the reviewers’ comments. The revised version has improved in clarity, methodological transparency, and overall organization.

Regards!

Reviewer #2: (No Response)

.

Reviewer #1: No

Reviewer #2: **Yes:** Mohammedamin HajureMohammedamin HajureMohammedamin HajureMohammedamin Hajure

---

## [Editor Report · Acceptance letter]

PONE-D-25-47275R1

PLOS One

Dear Dr. Kenea,

I'm pleased to inform you that your manuscript has been deemed suitable for publication in PLOS One. Congratulations! Your manuscript is now being handed over to our production team.

Kind regards,

on behalf of

A/Professor Zakir Abdu Adem

Academic Editor

PLOS One